# TRP Channels in Tumoral Processes Mediated by Oxidative Stress and Inflammation

**DOI:** 10.3390/antiox12071327

**Published:** 2023-06-23

**Authors:** Florentina Piciu, Mihaela Balas, Madalina Andreea Badea, Dana Cucu

**Affiliations:** 1Department of Anatomy, Animal Physiology and Biophysics (DAFAB), Faculty of Biology, University of Bucharest, 91-95 Splaiul Independentei, 050095 Bucharest, Romania; florentina.cojocaru@bio.unibuc.ro; 2Department of Biochemistry and Molecular Biology, Faculty of Biology, University of Bucharest, 91-95 Splaiul Independentei, 050095 Bucharest, Romania; mihaela.balas@bio.unibuc.ro (M.B.); madalina-andreea.badea@bio.unibuc.ro (M.A.B.); 3Research Institute of the University of Bucharest (ICUB), University of Bucharest, 90-92 Sos. Panduri, 050663 Bucharest, Romania

**Keywords:** ROS, TRP, inflammation

## Abstract

The channels from the superfamily of transient receptor potential (TRP) activated by reactive oxygen species (ROS) can be defined as redox channels. Those with the best exposure of the cysteine residues and, hence, the most sensitive to oxidative stress are TRPC4, TRPC5, TRPV1, TRPV4, and TRPA1, while others, such as TRPC3, TRPM2, and TRPM7, are indirectly activated by ROS. Furthermore, activation by ROS has different effects on the tumorigenic process: some TRP channels may, upon activation, stimulate proliferation, apoptosis, or migration of cancer cells, while others inhibit these processes, depending on the cancer type, tumoral microenvironment, and, finally, on the methods used for evaluation. Therefore, using these polymodal proteins as therapeutic targets is still an unmet need, despite their draggability and modulation by simple and mostly unharmful compounds. This review intended to create some cellular models of the interaction between oxidative stress, TRP channels, and inflammation. Although somewhat crosstalk between the three actors was rather theoretical, we intended to gather the recently published data and proposed pathways of cancer inhibition using modulators of TRP proteins, hoping that the experimental data corroborated clinical information may finally bring the results from the bench to the bedside.

## 1. Introduction

Transient receptor potential (TRP) channels constitute a superfamily of cation-permeable (Na^+^, K^+^, Ca^2+^) ion channels that exhibit extraordinary diversity in structure, activation mechanisms, and physiological functions. Today, 30 years after discovering the first TRP channel, this superfamily includes more than 100 members, in vertebrates and invertebrates and grouped into seven families based on structural homology: TRPC (classical or canonical), TRPV (vanilloid), TRPM (melastatin), TRPA (ankyrin), TRPML (mucolipin), TRPP (polycystin), and TRPN (NOMPC—no mechanoreceptor potential C). Apart from TRPN proteins, which are found in invertebrates and fishes, all TRP proteins were identified in mammals (Drosophila TRP channels). In addition, many stimuli activate TRP channels such as temperature, membrane potential (voltage), oxidative stress, mechanical stimuli, pH, and ligands (endogenous and exogenous), thus illustrating their adaptability. Many TRP members are vital proteins in pain perception and integration of nociceptive signals; activating these TRP sensors leads to cellular depolarization by increasing the intracellular concentration of cations, resulting in protective responses such as pain or local inflammation [1].

As such, TRP channels modulate the functions of both excitable and non-excitable cells, mainly by calcium homeostasis [2]. The uptake of Ca^2+^ is specific for most TRP channels, with some members sharing a higher relative permeability to Ca^2+^ over Na^+^ (PCa/PNa). For example, TRPV5 and TRPV6 are specifically Ca^2+^-selective with PCa/PNa > 100 [3], whereas TRPM4 and TRPM5 have almost insignificant calcium permeability of <0.1 [4]. These differences come from the well-established structure of the selectivity filter and the variability of the amino acids lining the central pore. All TRP family members generally form tetramers, and each subunit shares the same structure: six transmembrane domains with the pore region between the fifth and the sixth domain. The contribution of TRP channels in so different cellular functions relies on the pore structure and their gating properties, modulated by various stimuli. Additionally, detailed structural models predicted how endo- and exogenous factors regulate these complex proteins.

This review envisaged the role of TRP channels that are sensors for oxidative reactions in cancer cells. In recent years, the specific TRPs directly or indirectly activated by reactive oxygen species (ROS) were named redox-sensitive TRP channels [5]. We restrained this review to these TRP family members, and we focused on the work of the last ten years based on the intricate interplay between redox channels, inflammation, and cancer. Given the complexity of these processes addressed mainly in specific and independent studies, this review fills the gaps with possible mechanisms and signaling pathways.

## 2. Sources of ROS and Their Targets in Tumoral Cells

Oxidative stress represents the disequilibrium between generating and accumulating free radicals or peroxides. Free radicals are oxygens with an uneven number of electrons, which raises the capacity to interact with other molecules. In contrast, peroxides are compounds in which a covalent bond links two oxygen atoms. Free radicals, classified as reactive oxygen species (ROS), reactive nitrogen species (RNS), and reactive carbonyl species (RCS), are derived from exogenous and endogenous sources. Production of free radicals from exogenous sources may be consecutive to pollutant exposure, heavy metals, ingestion of some drugs, cigarette smoke, alcohol, or radiation. As a defense mechanism, cells rapidly release ROS producing an oxidative burst. Instead, tumoral processes, inflammation, infection, and exercise, are endogenous sources of free radical production [6]. In animal cells, ROS are formed primarily in mitochondria, peroxisomes, and in any other cellular compartment presenting proteins or molecules with high redox potential [7]. They are then removed or detoxified by antioxidative enzymes and antioxidants, thus helping the cells to maintain ROS at low concentrations.

The ROS with the higher cellular impact are hydrogen peroxide H_2_O_2_, superoxide O_2_^−^, hydroxyl radical OH^−^, and singlet oxygen O_2_. Generally, oxidative stress and free radicals are considered harmful to human organisms by breaking down DNA and proteins, but the generation of ROS is also beneficial, as recently reviewed in [8]. For instance, the immune system uses ROS to fight against pathogens. Studies document that macrophages use ROS to destroy invasive microorganisms, consecutive to oxidative burst mediated by NADPH [9]. The concept that ROS have physiological roles stay in the data proving that mitochondrial ROS improve skin development and regeneration [10], neuronal differentiation [11], or cell apoptosis [12]. However, most data support the deleterious effects of ROS, resulting from the interaction with the lipids’ membrane, proteins, or DNA.

ROS induce peroxidation of fatty acids and eventually impair membrane fluidity, protein–lipid interactions, and ion transport [13]. In cancer, lipid peroxidation triggers ferroptosis—the iron-dependent form of non-apoptotic cell death—consecutive to lipid hydroperoxide accumulation [14]. Ferroptosis occurs when the accumulation of lipid peroxides surpasses the cellular defense mechanisms. In addition to its effects on neurons, cardiac cells, and immune cells, recent studies reported the role of ferroptosis in cancer [15]. Enriching PUFA-PLs in cell membranes, the overload of iron, and the imbalance of ferroptosis defense mechanisms inhibit tumor growth and proliferation and even augment anti-tumor immunity. Indeed, previous data reported that immunotherapy-activated CD8+ T cells drive ferroptosis and suppress tumorigenesis after [16]. However, the relationship between the immune system and ferroptosis is more complicated than this, as shown by disputed results obtained from studies of interleukins and tumor necrosis factor-alpha (TNF-α). Although serum interleukin 6 (IL-6) is a mediator of inflammatory diseases and lung cancer [17] and indicates a worse prognosis [18], it unexpectedly promotes ferroptosis in bronchial epithelial cells [19]. Experiments on head and neck squamous cell carcinoma and clinical data correlate ferroptosis resistance with tumor progression [20,21].

Lipid peroxidation generates short-chain aldehydes and malondialdehydes (MDAs), which, in turn, inactivates proteins that promote ferroptosis. A type of toxic aldehyde is 4-hydroxynonenal (4-HNE), which can induce structural and functional changes by the covalent binding to different proteins. In contrast to other unstable ROS, 4-HNE is a lipid peroxidation product with a high affinity for proteins forming stable complexes. Additionally, 4-HNE strongly influences the genome and affected cells’ differentiation, proliferation, and apoptosis [22]. Specifically, 4-HNE reacts with thiol groups, forming covalent bonds, and impacts posttranslational modification of proteins.

As mentioned above, lipid peroxidation alters the bilayer structure and membrane fluidity [23], whereas lipid composition modulates ROS production. For example, reducing free cholesterol in membrane microdomains named lipid rafts diminishes ROS production [24]. Moreover, many studies showed that TRP channels localize within lipid rafts [25] and in particular, lipid rafts modulate the members of the TRPC family [26].

ROS induce DNA damage and consecutively dysregulate cellular responses to DNA damage response (DDR). Among other inducers of DDR, anticancer agents such as the divalent platinum compound cisplatin (CDDP) and the anthracycline doxorubicin (Dox) increase ROS levels and modulate the expression of several TRP channels. The results comprise toxic effects or resistance to chemotherapies. Doxorubicin (Dox) intercalates into DNA, inhibits topoisomerase II, changes the chromatin structure, and increases ROS production. However, cardiomyopathy and symptomatic heart failure in up to 5% of cancer patients hamper its therapeutic application [27].

## 3. ROS and the Redox TRP Channels

As reported for redox TRP channels, recent studies showed that stimulation of TRPV4 induces exocytosis and leads to ferroptosis of human melanoma cell [28]. The authors showed that TRPV4 stimulation in melanoma cell lines changed mitochondria structure and crista morphology. These changes are specific to ferroptosis, experimentally assessed by changes in the expression of ferroptosis markers: iron metabolism-associated gene expression of FTH1 (ferritin heavy chain 1), ACSL4 (acyl-CoA synthetase long-chain family member 4), and PTGS2 (prostaglandin-endoperoxide synthase 2). When activated, TRPV4 triggers ferroptosis through signaling pathways linked to increased intracellular calcium levels and inhibits tumorigenesis. Another study proposed using TRPV4 agonist GSK1016790A in combination with cisplatin (an anticancer drug that produces ferroptosis) to treat oral squamous cell carcinoma [29]. However, Fujii S. et al. disputed these results. The authors proved that TRPV4 promotes cell proliferation [30] and reiterated the double-edged sword of TRP activation and ROS.

In a study on cardiac tissues collected from mice, the authors measured high levels of TRPA1 mRNA and protein after Dox treatment, correlated with myocardial lesions. The functional consequences were cardiac dysfunctions, attenuation of oxidative stress, increased proinflammatory cytokines, and endoplasmic reticulum stress [31].

TRPC1 expression predicts breast cancer vulnerability to Dox and pulsed electromagnetic field (PEMF) therapy [32]. The cumulative ROS production by combined PEMF and Dox therapy creates an oxidative environment sufficient to slow breast cancer cells growth. Furthermore, overexpression of TRPC1 increased the sensitivity to this therapy in animal models. These findings prompted the authors to propose the combinatory therapeutic strategy of PEMT + Dox, specifically for cancers characterized by high TRPC1 channel expression.

Another study reported that intensive use of doxorubicin induces pathological protein interaction of TRPC3 channels with NADPH oxidase 2 (Nox2) consecutive to cardiac atrophy in mice. Ibudilast, an anti-asthmatic drug, suppressed ROS production and cytotoxicity by destabilizing the TRPC3-Nox2 complexes, without reducing TRPC3 channel activity [33].

TRPM2 had protective roles in human breast adenocarcinoma cell lines, minimizing DNA damage. Furthermore, inhibition of TRPM2 with the specific antagonist, 2-APB, decreased cancer cell proliferation by about 60% and saved non-tumoral cells. These results indicate that using TRPM2, as a therapeutic target in breast cancer, can pave the way to a more effective treatment. Alternatively, tumors with lower TRPM2 levels may be more sensitive to chemotherapy [34].

Cisplatin is used to treat various solid cancers, but unfortunately, it has many different secondary effects, including hearing loss. The compound directly binds to DNA, accumulates in mitochondria, and generates ROS, which mediates DDR. Several studies showed that CCDP-induced ototoxicity is related to the production of 4-HNE. The authors also described Ca^2+^ overload in cochlear cells and overexpression of the TRPV1 channel. The secondary effects were reversed by ursolic acid (UA), a plant-derived triterpene compound with potent antioxidant effects and a blocker of TRPV1 [35].

Studies on HEK cells showed that 4-HNE inhibited TRPV1 currents and mediated Ca^2+^ influx. An in silico analysis revealed four putative binding sites in the pore helixes of TRPV1: cysteine residues 616, 621, and 634. However, mutational studies linked the effect to a key pore cysteine residue 621, responsible for the impaired activity of TRPV1 through direct binding of 4-HNE [36]. To understand the interaction of ROS products with different amino acid residues, we will briefly describe the redox TRPs structure in the next section.

## 4. The Structure of the Redox TRP Channels

The most common means by which ROS directly activate the TRP channels involves redox modification of the free thiol groups. Therefore, the channels with the best exposure of the cysteine residues and, consequently, the most sensitive to oxidative stress are TRPC4, TRPC5, TRPV1, TRPV4, and TRPA1 (Figure 1).

The TRPC family channels bear conserved cysteine residues situated in the pore-forming region and accessible from the cytoplasm denoted C549/C554 in TRPC4 and C553/C558 in TRPC5 [37].

Channels from the TRPV family have extracellular and intracellular cysteine residues. Oxidizing compounds activate rat and chicken TRPV1 by acting on extracellular Cys residues from the cytoplasmic side [38], whereas oxidizing agents activate human TRPV1 via dimerization. Using non-reducing SDS Page electrophysiology and mass spectrometry, Ogawa et al. suggested that from many Cys residues in the N terminal, Cys-127, -158, -258, -363, and -742 participate in intersubunit disulfide bond formation and are sensitive to oxidation. From these, the most sensitive residue assessed by half-maximum saturation of H_2_O_2_ was Cys 258 localized in the N-terminal domain of TRPV1 [39].

In a recent report, mutational studies revealed that cysteines from the extracellular domain of TRPV4 (C639, C645, C652, and C660) that share homology with TRPV1 cysteines are ROS sensitive [9]. Additionally, mutation of several intracellular cysteines (C294, C353, and C427) abolished the regulatory role of TRPV4 in the invadasome through a process directly linked to ROS sensing. Both extra and intracellular cysteine groups determine cell invasion related to the degradative functions of acto-adhesive and extracellular matrix [40].

TRPA1, the most ROS-sensitive channel, is activated either by extracellular compounds such as hydrogen peroxide (H_2_O_2_), hydroxyl radical (HO^•^), nitric oxide (NO), or by endogenously generated products such as 4-hydroxinonenal (4-HNE). This sensitivity comes from both cysteine residues (Cys 621, Cys 641, Cys 665) and, to a lesser extent, lysine 708, localized on their N-terminal segments [41].

We also define redox channels TRPC3, TRPM2, and TRPM7. TRPC3 may associate with TRPC4, forming a redox-sensitive channel in endothelial cells [42,43] and acting as a positive regulator of ROS in cardiomyocytes [44]. Additionally, cholesterol loading and phospholipase C stimulation increase surface expression of TRPC3. These experiments were performed in HEK cells stably expressing TRPC3 [45].

ROS indirectly activates TRPM2 and TRPM7 through ADP-ribose (ADPR) generated in the mitochondria by NADase hydrolyses [46]. A high-resolution structure of TRPM2 revealed that the N terminus contains four TRPM homology regions, while the C-terminus has a conserved coiled-coil tetramerization domain and a NUDT9-H domain. TRPM1/2 and NUDT9-H domains provide the binding sites for ADPR and other ADPR analogs, such as nicotinic acid adenine dinucleotide phosphate (NAADP) and deacetylation product release by sirtuins—OAADPr, which activate the channel [47].

TRPM7 is a channel negatively regulated by intracellular Mg^2+^. H_2_O_2_ inhibits TRPM7 currents in an Mg^2+^ and ATP-dependent manner [48]. Site-directed mutagenesis studies revealed that cysteine residues (C1809 and C1813) of the zinc-binding motif act as oxidative stress sensors and increase Mg^2+^ inhibition, perturbing the interactions between the channel domain and the kinase domain [49].

Apart from cysteines, free radicals may target methionine from TRP ion channels, which turns to methionine sulfoxide and alters the protein structure and, consequently, channel gating and signaling pathways. For instance, mutagenesis analysis showed that the redox sensitivity of TRPV2 depends on methionine residues M528 and M607 [50].

The oxidation of the sensitive residues of redox TRP channels has various effects on the tumorigenic process and modulates the proliferation and growth of cancer cells, migration, and apoptosis, depending on the ROS sources and their interaction with different cellular compartments.

Other studies sustained the use of agonists and antagonists of redox TRP channels for their effects on proliferation, tumor growth, or cell apoptosis. However, it is only sometimes clear whether these compounds are efficient because of the interaction with channels pore or by a direct action on other cellular compartments. In the following subchapters, we review recent data on the participation of redox TRPs in tumorigenesis upon the activation by ROS.

## 5. Regulation of the Redox TRP Channels by Oxidative Stress in Cancer Cell Proliferation

The fine tune of ROS levels affects cancer cell proliferation and uses different pathways. Most of the results related to redox TRP come from experiments showing that TRPM2 and TRPM7, indirectly activated by ROS, influence cell proliferation. Figure 2 summarizes the results presenting the signaling pathways activated in cell proliferation by TRPM2 and TRPM7.

Firstly, we discuss the compelling evidence coming from the analysis of TRPM2 in acute myeloid leukemia (AML). Using CRISPR/Cas9 technology, the results showed that in TRPM2-depleted leukemia cells, ROS levels were higher, while mitochondria function diminished. In these conditions, cell proliferation significantly decreased. The authors bring strong evidence that TRPM2 regulates mitochondria bioenergetics, cell proliferation, and autophagy in AML by synchronous modulation of several transcription factors such as HIF-1/2α, Nrf2, ATF4, and CREB [51]. Using the same technology, the study of Bao L et al. [52] suggests that TRPM2 preserves survival of neuroblastoma cell lines and xenografts after oxidative injury. Here, the authors revealed the importance of the nuclear factor erythroid 2-related factor (Nrf2).

Additionally, TRPM2 regulates the tumorigenic process by activating major signaling pathways: JNK, PKC, or MAPK. The effect may depend on the splice variant expressed in different tumors. For instance, the isoform, TRPM2-S, lacking the four C-terminal transmembrane domains and the putative Ca^2+^ pore, is the prognostic biomarker for retroperitoneal liposarcoma (RPLS), which modulated proliferation and apoptosis via ERK and AKT pathways. Under oxidative stress conditions, mimicked by high levels of H_2_O_2_, the TRPM2-S enhanced apoptosis. In normoxia conditions, this isoform promoted the proliferation of RPLS cells [53].

Xing, Y. et al. systematically explored the role of TRPM7 channels and various modulators (Clozapine, Naltriben, Proadifen, Mibefradil, FTY720, and NS8593) on several cancer cell lines, in vivo xenograft, and metastasis mouse models. The study showed that pharmacological activation of TRPM7 inhibits autophagy and suppresses proliferation, as quantified by changes in microtubule associated protein 1 light chain 3 (LC3) and p62. The TRPM7 agonists Clozapine and Naltriben elevated ROS levels, while TRPM7 silencing inhibited ROS production by impairing mitochondrial turnover [54].

Missing conspicuous data of a direct liaison between activated ROS TRPA1/V1 and proliferation, we propose a possible link that emerged from studies that independently analyzed ROS’s modulation or the role of TRPA1/V1 on cell proliferation.

Experiments performed on lung adenocarcinoma cells proposed a bold hypothesis: the N-terminal ankyrin repeats of TRPA1 directly bind to the C-terminal proline-rich motif of the fibroblast growth factor receptor 2 (FGFR2), inducing the activation of the receptor through the MAP kinase pathway [55]. TRPA1 is the primary ROS sensor, and the ankyrin repeats may form protein scaffolds with unexpected functions in different tumoral cells. However, no other studies support these findings, and TRPA1 remains a proposed target mainly based on the uptake of Ca^2+^. TRPA1 regulates intracellular Ca^2+^ levels and the ryanodine receptor (RyR) in response to oxidative stress. Consecutive to the Ras/MAPK pathway activation, both proteins trigger intestinal stem cell (ISC) proliferation [56].

In lung adenocarcinoma, the expression of TRPA1 and TRPV1 were upregulated by the hypoxia-inducible factor 1α (HIF1α) nuclear accumulation, finally leading to cell proliferation. The study proposes that the signaling pathway TRPV1-Ca^2+^-nNOS-NO activates proliferation and metastasis in these tumoral cells [57]. In other instances, TRPV1 hinders cell proliferation. For example, in skin carcinoma cells, pancreatic adenocarcinoma, or human colorectal carcinoma, TRPV1 suppresses posttranslational modification of the epidermal growth factor receptor (EGFR) associated with pro-proliferative pathways [58]. The results presented so far sustain the role of TRPA1 and TRPV1 in cancer cell proliferation but are irrelevant to the modulation of channels by ROS. Data obtained from studies unrelated to ion channels prove without doubt that ROS modulate cell proliferation [59]. A possible, so far missing from experimental evidence, link between TRPA1, ROS, and cell proliferation is feasible because the main endogenous activators for TRPA1 in cancer are the products of oxidative stress. Figure 2 illustrates a diagram of this hypothesis.

In cancer cells, proliferation is intimately regulated by the apoptosis of neighboring cells through a process defined as “apoptosis-induced proliferation (AiP)” [60]. Moreover, apoptotic caspases mediate the generation of ROS for promoting AiP [61]. Additionally, TRP channels were indicated as important regulators of apoptosis, especially by their function in Ca^2+^ homeostasis [62]. Therefore, we will cover the crosstalk TRP–ROS-apoptosis in the following subchapter.

## 6. Apoptosis Induced by Modulation of the Redox TRP Channels

Calcium is an essential second messenger in the process of tumorigenesis, participating in migration proliferation, apoptosis, and cell cycle progression. The canonical transient receptor potential channels (TRPCs) contribute to the protein tyrosine kinase- or G protein-coupled receptor-operated Ca^2+^ entry (ROCE) and Ca^2+^ store-operated Ca^2+^ entry (SOCE), being essential players in signaling pathways involving calcium. A recent study showed that TRPC3 is upregulated in gastric cancer (GC) patients and cell lines [63]. Furthermore, the specific antagonist Pyr3 (a pyrazole compound) increased early and late apoptotic and necrotic GC cells, whereas TRPC3 knowdown displayed a significant decrease in ROS levels compared to wild-type (WT) cells. These results suggest that TRPC3 protects against apoptosis and facilitates ROS production in GC cells. Moreover, TRPC3 upregulation correlates with CNB2/GSK3β/NFATc2 pathway activation in GC issues.

In breast cancer cells, TRPA1 suppressed apoptosis upon stimulation by H_2_O_2_, whereas TRPA1 inhibition increased cell apoptosis in response to platinum-based drugs [64]. Interestingly, while inhibition of TRPA1 was determinant for apoptosis, channel silencing had no effect. These results prove that apoptosis induced by TRPA1 is related to Ca^2+^ signaling and to channel conducting properties of TRPA1. However, in pancreatic adenocarcinoma cells, the process is different: downregulating TRPA1 with siRNA, decreased cellular viability, and possibly, apoptosis, as revealed by cell cycle analysis [65]. These demonstrated that TRPA1 expression is important in the proliferation and apoptosis of pancreatic ductal adenocarcinoma cells without activation with specific agonists. Furthermore, the activation of the transcription factor Nrf2 using H_2_O_2_-enhanced TRPA1 expression, which occurs only in serum-starved conditions.

In glioma cells (GMB), the standard of care is surgery followed by external-beam radiation with concomitant temozolomide (TMZ) therapy. TMZ produces cell apoptosis and increases ROS production, but the treatment has short-term results due to chemoresistance. A recent study showed that activating TRPA1 in glioma cells leads to mitochondrial damage and cell apoptosis and decreases TMZ resistance. This process relies on increased ROS levels and apoptosis after the pretreatment of GMB cells with a TRPA1 agonist [66].

TRPV1 expression or functional modulation by agonists and antagonists affects apoptosis in many cancer cells, as carefully revised in [58]. For instance, TRPV1 regulates cell proliferation in different tissues, including bone-derived tumor cells inducing cell proliferation in osteoblasts and cell apoptosis in osteoclasts lacking the TRPV1 [67] and mineralization. TRPV1 is also activated by zoledronic acid, an effective anticancer drug in bone metastatic disorders [68]. Several studies monitored ROS production and its relation to TRPV1. This included reports from breast cancer cells, where a modulator of the TRPV1 channel decreased cell viability and increased apoptosis and ROS production [69]. These results favor the hypothesis that an allosteric modulator may be a better putative therapeutic agent than the classical TRPV1 activator capsaicin, which induces pain when applied in high and repeated doses. The agonists and the antagonists of TRPV1 raised the interest of many groups in cancer therapy. A study performed on different cancer cell lines showed that capsazepine, a TRPV1 antagonist, upregulates proapoptotic proteins p53 and Bax, which augments TRAIL-induced apoptosis. Finally, the study suggests that the induction of TRAIL receptors activates the ROS–JNK–CHOP pathway [70].

Overexpression of TRPV4 in lung cancer cells promotes apoptosis by stimulating the p38 MAPK pathways [71]. In squamous cell carcinoma (SSC), oxidative stress induced apoptosis within 24 h of H_2_O_2_ application. Data indicated that TRPM2 was activated and promoted apoptosis via caspases-3 and -9 in cultured SCC cells [72]. In irradiated human T leukemia cells, TRPM2 was denoted as a “death channel” because it promoted apoptosis and G2M cell cycle arrest through autophosphorylation and activation of Ca^2+^/calmodulin-dependent protein kinase II (CaMKII) [73]. Because ionizing radiation confers direct and indirect oxidative stress, the study hints toward crosstalk between TRPM2, ROS, and apoptosis.

Additionally, clostridium botulinum neurotoxin A (BTX) inhibits proliferation, growth and induces apoptosis in neuroblastoma and glioblastoma cells. The underlying mechanism may be the upregulation of mitochondrial oxidative stress and activation of the TRPM2 channel [74]. Together, these results indicate a clear liaison of TRP redox channels with apoptosis (Figure 3) upon modulation by ROS, intuitively via the Ca^2+^ entry, which is a clear activator of apoptosis.

## 7. TRP Channels Modulation by Oxidative Stress in Cancer Cell Migration

Many studies reported that ROS modulate cell motility, and different reports suggest identical effects of several TRP channels. The evidence of a physical link or a common signaling pathway is unclear, but we review and explain these putative interactions and condense them in Figure 4, which schematically illustrates a migratory cancer cell.

Some data stand that ROS inhibits cancer cell migration. For example, a study performed on mice with malignant melanoma and on several cell lines showed that the antioxidant N-acetylcysteine (NAC) increased the number of lymph nodes, as well as the proliferation and migration of tumor cells by boosting GSH concentrations and activating RhoA [75]. These results indicate that ROS inhibit cell proliferation and migration in these tumors. Of course, one may argue that melanocytes sequester ROS [76] preventing a significant increase of transcription factors sensitive to ROS, therefore, the effect of NAC is elusive. Nevertheless, other experiments support the idea and qualify ROS as friends rather than foes, reducing the migration of tumoral cells. For instance, studies on liver cancer cells showed that H_2_O_2_ inhibited migration via the DLC1/RhoA signaling pathway [77] (pathway 1 in Figure 4). Additionally, in advanced gastric cancer cells, H_2_O_2_ inhibited migration by modulation of the PI3K/Akt signaling pathway [78].

The signaling pathway of cell migration by ROS came from a recent study that evaluated cell–cell adhesion of head and neck squamous cell carcinoma [79]. The downstream effectors of ROS activation are Src kinases and STAT3 transcription factors (pathway 2 in Figure 4). Along with protein kinase C (PKC), protein kinase A (PKA), and CaM-Kinase II, Src phosphorylates many TRP channels [80]. An early study showed that Src kinases phosphorylate TRPV4 during hypotonic stress [81] and, recently, TRPV4 was associated with the migration of endometrial cancer or melanoma cells [82,83]. TRPV4 induced melanoma metastasis by Src-cofilin intracellular pathway but, unexpectedly, this study showed that TRPV4 activates the phosphorylation of Src kinase. This is a distinct circumstance where a TRP channel activates the phosphorylation of other proteins downstream the migratory pathway. TRPM7 and TRPM6 are also able to phosphorylate other proteins because they contain kinase domains [84]. Indeed, TRPM7 inhibition suppressed cell migration and the phosphorylation of Src, Akt, and JNK (c-Jun N-terminal kinase) kinases in bladder cancer cells [85]. The auto-phosphorylation of these channels possibly contributes to the migration in specific tumoral environments [86], but data do not yet support this idea. Two hypothetical models emerged from the data presented so far. One implied that ROS activates Src, which phosphorylates TRPV4 and promotes cell motility. The other may state that ROS activated TRPV4/M7, which phosphorylates other kinases will mobilize proteins for cell migration and eventually metastases. In these cases, blocking channel phosphorylation will impact cell migration, but these theories are yet to be tested.

Less speculative is the relation of TRPM2 phosphorylation with ROS (pathway 3 in Figure 4). H_2_O_2_ activates and induces phosphorylation of TRPM2 and the non-specific inhibitor of Src family kinases PP2 blocks the effect [87]. Independent of phosphorylation, TRPM2 promotes the migration of gastric cancer cells, through a pathway related to Ca^2+^ entry and regulation of the PTEN/Akt pathway [88]. In neuroblastoma, the same pathway is activated by TRPM2, along with the transcription factors HIF-1α1, E2F1, FOXM1, and CREB. Activation of TRPM2 increased the expression of α1, αv, β1, and β5 integrins and, consecutively, migration and invasion of neuroblastoma [89].

Src phosphorylates another member of the TRPM family, TRPM8, and Src inhibition downregulates TRPM8-mediated responses [90]. Most importantly, Src- regulates the inflammatory NF-κB signaling pathway and 4-HNE activates it. In this scenario, we may picture a better image of oxidative stress and inflammation, as we can add the putative roles of TRP channels in this signaling pathway.

## 8. Redox TRP Channels and Inflammation

Studies have long established the physiological role of inflammatory molecules in cancer. Lately, the tumor microenvironment was declared the main culprit for the inflammatory response in neoplasms [91]. In many cancers, the tumor is invaded by macrophages, T cells, myeloid cells, natural killers, and others that secrete cytokines (e.g., TNF-α and IL- cytokines) and active mediators, including ROS [92]. The relationship between ROS and inflammation is again dual. On the one hand, ROS activate the NF-κB and JAK-STAT signaling pathways, which are positive regulators of cancer cells [93]. On the other hand, NF-κB decreased tumor development in some cancers. For instance, a model of hepatocellular carcinoma (HCC) induced in mice by the tumor initiator diethylnitrosamine (DEN), prevented NF-kB-induced hepatocarcinogenesis [94]. Additionally, in human epidermal cells, both NF-κB and oncogenic Ras trigger cell-cycle arrest [95].

In the same line, activation of redox TRP channels may have dual roles in cancers associated with the inflammatory process because they activate different signaling pathways depending on the tumor type. Moreover, anti-inflammatory compounds modulate redox TRPs differently (Table 1). Therefore, a better insight into the use of anti-inflammatory drugs in cancer may come from understanding the modulation of TRP channels and the effects on the tumorigenesis process.

Of all members of the TRP superfamily, TRPA1 was named “the gatekeeper for inflammation” [106]. Activation of TRPA1 in glioma cells significantly decreased antioxidant expression and increased mitochondrial ROS production. The effects were reversed by pretreatment of glioma cells with the TRPA1 antagonist, as opposed to the temozolomide (TMZ) therapy, which had a lower impact on the production of mitochondrial ROS and cell apoptosis [66]. The authors proposed a scenario in which GBM cells develop TMZ resistance because of ROS production and consecutive TRPA1 activation. From this study, TRPA1 emerged as a tumor-suppressor protein and its activation may be helpful for better therapy. In agreement with these results, TRPA1 decreased cell proliferation and migration in pancreatic adenocarcinoma cells [65]. Opposite results were obtained from breast cancer cells, where inhibition of TRPA1 reduced tumor growth [107]. Additionally, melanoma cells developed oxidative stress defense mechanisms upon TRPA1 activation [108].

The latest research on TRPA1 in inflammatory diseases and the signaling pathways were exhaustively reviewed in [109]. Although the authors skipped the relation of inflammation with cancer, they presented several cellular pathways. For instance, they mentioned that activation of TRPA1 reduces the expression of proinflammatory cytokines through the nuclear factor kappa-B (NF-kB) signaling pathway, described in a previous study by Lee KI et al. [110]. Nevertheless, cancer pain is associated with TRPA1 blockade by triterpenoids or riterpenoids ([111] and channel activation by ROS, but the process takes place in nociceptors and not in tumoral cells.

Another redox channel, TRPV1, activated the inflammatory pathway Nf-kB and STAT3. Convincing evidence comes from a study on knockout mice for TRPV1, which develop colon tumors, through a mechanism tightly related to inflammation [112]. Additionally, proinflammatory (IL-1 and IL-6) and invasion factors (MMP9) activated these signaling pathways.

The oxidative stress by-product, 4-HNE, is linked to inflammatory bowel disease (IBD) and colorectal cancer (CRC). High concentrations of 4-HNE were measured in DSS-induced IBD in C57BL/6 mice, by the activating Toll-like receptor 4 (TLR4) signaling pathway [113]. In addition to impairing the quality of life, IBD increases the risk of CRC, associated with severe oxidative stress and high 4-HNE levels [114]. Then, again, redox TRP channels may be related to 4-HNE production and inflammation in CRC, working either as oncogenic factors or as tumor suppressors. For instance, high expression levels of TRPC5 in colon cancer patients correlate with poor prognosis and metastasis, via the HIF-1α/Twist signaling pathway [115]. Additionally, we know from studies on hippocampal neurons that TRPC5 inhibition with a specific antagonist significantly reduced i extracellular 4-HNE concentration [116]. Putting together these data, we may speculate that inhibition of TRPV5 reduces 4-HNE and could impair metastasis of colon cancer cells.

Many studies indicated TRPV1 as a tumor suppressor of CRC; therefore, its activation may be feasible for this neoplasm [117]. We also know that 4-HNE inhibited TRPV1 Ca^2+^ current in HEK transfected cells [36], encouraging the use of the antagonist capsazepine, as a pharmacological tool for CRC.

A type of CRC, so-called colitis-associated cancer (CAC), is one of the complications of long-standing inflammatory bowel disease, characterized by the massive production of ROS and inflammatory cytokines that activate signaling pathways, such as NF-κB, STAT3, p38 MAPK, and Wnt/β-catenin, which, in turn, modulate the tumorigenic process [118]. A previous study investigated the pathogenesis of CAC in mice [119]. It reported that TRPV4 expression is upregulated in CAC and contributes to the progression of colonic inflammation and vascular permeability. Therefore, the authors propose the inhibition of TRPV4 as a therapeutic strategy for CAC, which may decrease cytokines production and tumorigenesis.

Previous studies measured high expression levels of TRPA1 and TRPM2 in mice with induced CRC and a direct relationship with expression levels of TNF-α and Casp-3 [120]. Therefore, the authors propose pomegranate extract as modulator of these proteins.

Oxidative stress and 4-HNE are the driving forces for esophageal squamous cell carcinoma (ESCC). In a cohort of three non-malignant esophageal epithelial tissues, 11 esophageal in situ carcinoma tissues, and 57 ESCC tissues from patients in the Chaoshan area, high levels of 4-HNE `correlated to the degree of inflammation in ESCC [121]. In addition, in samples from patients with ESCC, TRPM7 was found as an independent prognostic factor of good postoperative survival [122].

Contrary to these results, a retrospective study of oropharyngeal squamous cell carcinoma (SCC) in 102 patients showed the presence of 4-HNE in tumor cells, but a higher HNE-immunopositivity in non-malignant cells that were localized close to the carcinoma. The study concluded that 4-HNE is a bioactive marker of oxidative stress, but with defensive, anticancer potential of the surrounding non-malignant cells [123]. Similarly, TRPM2 has dual roles in SCC cell lines [61]. On the one hand, high plasma membrane TRPM2 expression protects against the early stages of cell growth. On the other hand, increased levels of nuclear TRPM2 during the advanced stages of cancer may stimulate tumorigenesis. Modulating TRPM2 by agonists and antagonists would be a rather tedious mission in this case.

If the inflammatory response depends on the tumor microenvironment (TME), it is equally valid that the imbalance of ROS shapes TME. Moreover, growth factors, cytokines, and inflammatory factors released by TME cells shape the fate of the tumor. First, it is essential to quantify the degree of infiltration of immune and stromal cells in tumors using the ESTIMATE algorithm to calculate immune and stromal scores. A recent study reported a significant relationship between members of the TRP superfamily, ESTIMATE scores, and immune-activation-related genes in almost 33 types of cancer study [124]. This pan-cancer analysis disregarded ROS implication in the channel activation, but among the TRPs correlated to TME, TRPC4 and TRPV4 are redox channels; therefore, ROS may be the endogenous activator in stromal and immune cells.

Senescence in TME has tumor-suppressive and tumor-promoting roles and correlates to therapy resistance [125]. A recent study showed that senescence depends on TRPC3 expression in the endoplasmic reticulum (ER), which physically interacts with the inositol 1,4,5-trisphosphate receptors (IP3R) and constitutively inhibits IP3R Ca^2+^ release [126]. Alteration of Ca^2+^ homeostasis induces secretion of several soluble factors, referred to as senescence-associated secretory phenotype with pro-tumoral effects.

## 9. Conclusions

Many recent investigations on human tumors and cell lines showed high expression of redox TRP channels, but as shown here, these polymodal proteins have dual roles as suppressors or initiators of the tumoral process. Because many cancer types show resistance to radio- or chemotherapy, the main objective of studies on TRP proteins is finding new targets and understanding signaling pathways that will meet the need for new therapeutic agents. In this context, the draggability of TRP channels and their modulation by compounds that interfere with tumorigenesis, such as ROS, appeal to both the scientific community and pharmaceutical companies. Additionally, the agonists and antagonists of TRP, often natural compounds that avoid radioactive flux, could be beneficial. We reviewed here studies that captured the relationship between TRP and ROS, TRP and inflammation, and we tried to make a link that explains the crosstalk between these actors in the tumorigenic process.

## Figures and Tables

**Figure 1 antioxidants-12-01327-f001:**
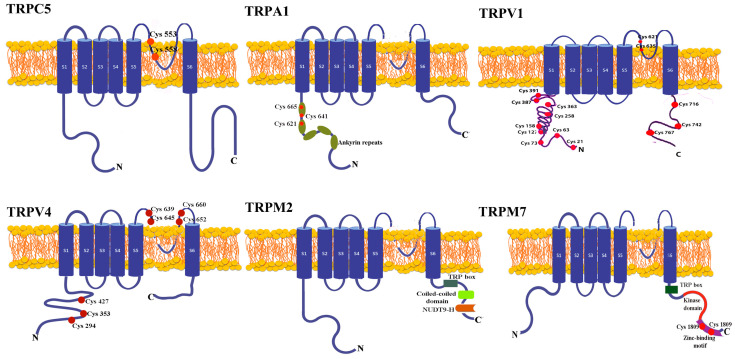
The topological structure of redox TRP channels. Each subunit comprises six transmembrane domains (S1–S6), one loop between S5 and S6 containing the channel pore, N and C termini located intracellularly. Illustrated in red are the cysteine residues exposed to oxidation. TRPC4/TRPC5 have two conserved cysteine residues in the pore-forming region. TRPV1 and TRPV4 have cysteine residues intracellularly and extracellularly located. TRPA1 contains ankyrin repeats, while TRPM2 and TRPM7 have a conserved TRP box. The coiled-coil region and NUDT9-H domain from TRPM2 provide the binding site for ADPR. TRPM7 has a zinc-finger domain containing two cysteines exposed to oxidation.

**Figure 2 antioxidants-12-01327-f002:**
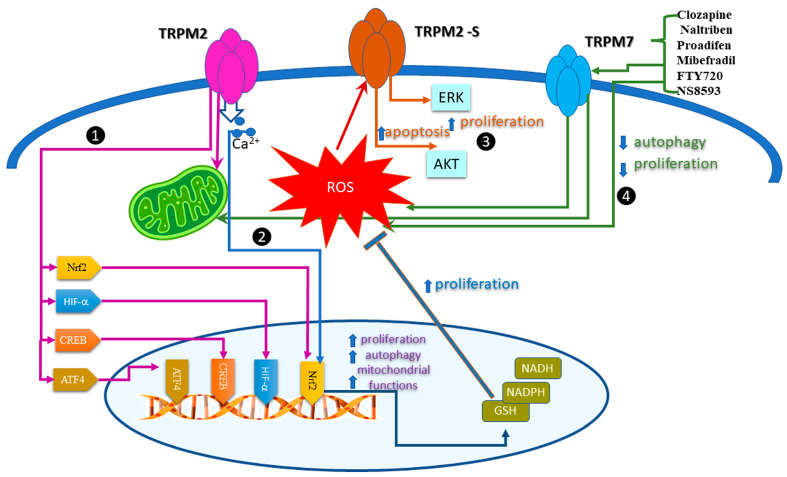
Signaling pathway activated by redox TRP channels in cancer cell proliferation. (1) TRPM2 activates mitochondria and transcription factors: HIF-1/2α, Nrf2, ATF4, and CREB; induces acute myeloid leukemia (AML) cells proliferation. Magenta connectors indicate this pathway. (2) Illustrated in blue connectors, this pathway shows that Ca^2+^ uptake by TRPM2 promotes survival of neuroblastoma cells. Nrf2 activates the antioxidant response and cofactors GSH, NADPH, and NADH. (3) ROS activates TRPM2-S and consecutively increases apoptosis and decreases proliferation in retroperitoneal liposarcoma (RPLS) cells, through AKT/MAPK pathways as depicted with scarlet connectors. (4) Agonists of TRPM7 stimulate ROS production and minimize mitochondrial turnover. The connections between TRPM7, ROS production, and consecutive reduction in autophagy and proliferation are displayed with green connectors.

**Figure 3 antioxidants-12-01327-f003:**
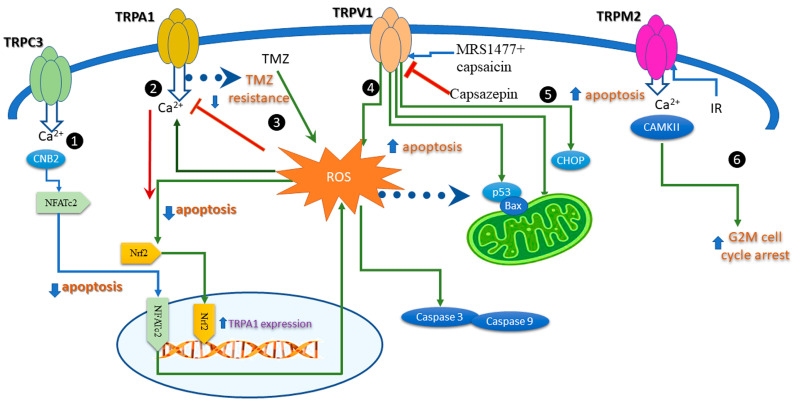
Intracellular pathways activated by redox TRP channels triggering apoptosis in cancer cells. (1) TRPC3 stimulates nuclear translocation of the nuclear factor of activated T cell 2 (NFATc2) by calcineurin Ca^2+^-binding subunits (CNB2) signaling, and consecutively activates ROS. (2) Calcium entry through TRPA1 inhibited apoptosis upon ROS activation. The process involves the nuclear factor erythroid 2-related factor (Nrf2). The model is valid for breast cancer cells. (3) Temozolomide (TMZ) activates ROS, TRPA1, and apoptosis in glioblastoma. The process is related to TMZ resistance. (4) Concomitant application of TRPV1 agonists (MRS1477 and capsaicin) increased apoptosis and ROS production. (5) Capsazepine, a TRPV1 antagonist, upregulates proapoptotic proteins p53 and Bax, which augment TRAIL-induced apoptosis. The activation of TRAIL receptors triggers the ROS–JNK–CHOP pathway. (6) after irradiation (IR), TRPM2 promotes apoptosis and G2M cell cycle arrest in leukemia T cells.

**Figure 4 antioxidants-12-01327-f004:**
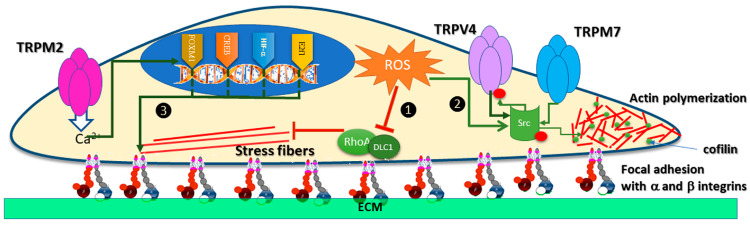
TRP redox channels modulate the migration of cancer cells. (1) ROS inhibits stress fibers and consecutively migration through the downregulating the DLC1/RhoA pathway. (2) Migration mediated by Src kinases. ROS stimulates Src, which in turn phosphorylates TRPV4 and TRPV7. TRPV4 channel contributes to migration via Src-cofilin intracellular pathway, while the TRPV7 kinase domain modulates Src phosphorylation. (3) Ca^2+^ entry via TRPM2 activates transcription factors HIF-1α1, E2F1, FOXM1, and CREB, which modulate the expression of α and β integrins. The proteins α and β integrin from the focal adhesion complex engage the extracellular matrix (ECM) in cell migration.

**Table 1 antioxidants-12-01327-t001:** Anti-inflammatory drugs with effects on tumorigenesis and modulators of TRP channels.

Anti-Inflammation Drug	TRP Channel Affected	Role in Tumorigenesis	Ref.
Nonsteroidal anti-inflammatory drugs (NSAIDs), e.g., aspirin and cyclooxygenase	Inhibition of TRPC4/C5	reduces the risk of breast cancer	[48,52,96]
Etodolac (Cox-2 inhibitor)	Desensitization of TRPA1	Inhibition of prostate and colorectal carcinoma cells proliferation	[97,98]
Cannabinoids and Cannabis Extracts	Activation of TRPV4 and TRPA1	Inhibits growth of breast cancer cells	[99,100]
Methotrexate	Inhibition of TRPM2	Chemotherapy of breast cancer, head, and neck cancer, lymphoma	[101]
Capsaicin	Activation of TRPV1	Senescence and Apoptosis in cervical cancer cells	[102,103,104]
Capsazepine	Inhibition of TRPV1	Inhibits growth and survival of prostate cancer, breast cancer, colorectal cancer, oral cancer, and osteosarcoma	[105]

## Data Availability

Not applicable.

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
