# Peer review of "TRP Channels in Tumoral Processes Mediated by Oxidative Stress and Inflammation"

_antioxidants, 2023, doi:10.3390/antiox12071327_

Round 1
Reviewer 1 Report
The review deals with the role of the TRP channel family in the redox state and inflammation in cell proliferation which is a relevant but not novel topic.
Line 65 The objective of the review: "A key question is whether redox channels can serve as therapeutic targets or prognostic markers for cancer. To address this problem, we will compare studies proposing means to use TRP redox channels in cancer therapy and emit the addressability of these complex proteins as tumoral biomarkers" that are not satisfied in the presented work. The reported data are limited to cell studies and no omics/transcriptomic data are reported. In my opinion, the aim should be limited to the role of the TRP and the redox state in cell signaling as it is.
The references do not fully report all significant data, see for instance what needs to be considered:
Line 497: TRPV1 regulates cell proliferation in different tissues including bone-derived tumor cells inducing cell proliferation in osteoblasts and cell apoptosis in osteoclasts lacking the TRPV1 (Scala et al., 2019) and mineralization. TRPV1 is also activated by zoledronic acid, an effective anticancer drug in bone metastatic disorders (Scala et al., 2019, 2022).
Scala R, . Zoledronic Acid Modulation of TRPV1 Channel Currents in Osteoblast Cell Line and Native Rat and Mouse Bone Marrow-Derived Osteoblasts: Cell Proliferation and Mineralization Effect. Cancers (Basel). 2019 Feb 11;11(2):206. doi: 10.3390/cancers11020206.
Scala R Bisphosphonates Targeting Ion Channels and Musculoskeletal Effects. Front Pharmacol. 2022 Mar 15;13:837534. doi: 10.3389/fphar.2022.837534. eCollection 2022.
Editorial
Line 339 in…
Line 347. Punctuation
Reviewer 2 Report
General comments
Picu et al. present here an overview on the potential role of the TRP ion channel family in tumorigenesis and inflammation. The authors provide initially an outline of the current mechanistic understanding of redox sensing by TRP channels as well as of their potential role in redox homeostasis. Furthermore, the potential role of redox associated TRP signaling in tumor growth and in anti-tumor defense mechanisms as well as the potential value of TRP channels as anticancer drug targets is discussed. The authors thereby address a timely and interesting topic. Unfortunately, the review manuscript is not well structured and the organization of the review is, in some parts, puzzling for the reader. Moreover, a few mechanistic and conceptual aspects need more attention as outlined below.
Specific comments:
1) Already the title of the review is slightly puzzling: What is exactly meant by: “friend or foes”? TRP channels and tumoral processes?….or TRP channels and redox stress?….
The authors may wish to rephrase the title of the review to make the message clear.
2) In the initial mechanistic part, the authors make the important discrimination between direct and indirect cellular regulation of TRP channels by redox stress. It is pointed out that indirect regulation involves redox modification of membrane lipids. Indirect modification has early on been shown also for TRPC3, which was reported as a redox sensitive channel ( Balzer et al. 1999, doi: 10.1016/s0008-6363(99)00025-5.; Poteser et al 2006, doi: 10.1074/jbc.M512205200) presumably based on its lipid, especially cholesterol sensitivity and/or targeting to lipid rafts. The aspect of redox stress-induced modification of membrane cholesterol as a crucial mechanism affecting plasma membrane architectures and thereby TRP channel function needs some/more attention.
3) The overall organization of the review paper need some correction. E.g. in #3 “sources of oxidative stress….” The authors combine mechanistic aspects of channel regulation with pathological aspects of redox stress as well as aspects of tumor pathology. This section is highly confusion and needs amendment. It includes also a duplication of the headline #3.1/3.2 “Role of TRP channels in ROS-induced DNA damage “. This topic appears actually not in focus of 3.1 at all! Mechanisitc and pathophysiological aspect should be better separated.
4) The role of TRP channel redox sensitivity in inflammatory processes as an aspect of cancer is certainly a very important issue in the review. The authors initially, in #5 mention that this aspect relates to alterations in the tumor microenvironment (TME). However, this rather novel key concept of tumor biology is not further addressed in more detail. How would TRPs and their redox sensitivity impact on the TME?
The manuscript needs some additional style editing
Round 2
Reviewer 1 Report
The manuscript is improved
The manuscript is improved
Author Response
We thank the reviewer for carefully reading and accepting the new version of the manuscript.
Reviewer 2 Report
The authors have indeed done a good job to amend this review paper. As it stands now, the paper is definitely a valuable overview.
I have only one minor remark/criticism: Changes in channel expression as well as surface presentation are essential mechanism of cellular regulation, including regulation by redox stress. These two mechanisms should not be mixed up as e.g. for the effects of cholesterol on TRPC3 (lines 214/215) - here PLC activation and cholesterl change mainly surface "expression" (presentation in the plasmamembrane). This should be emphasized/corrected. Also the term "HEK-transfected cells" is inapropriate and need rewording or should be simply deleted - it means ..HEK cells transfected with....
Author Response
We thank the reviewer for accepting the new version of the manuscript and also for the careful reading, which considerable improved our work.
We corrected the phrase as indicated and changed it to: " Also, cholesterol loading and phospholipase C stimulation increase surface expression of TRPC3. These experiments were performed in HEK cells stably expressing TRPC3 [45].
We also added the reference that we missed in the previous version.